# Quantum metrology with quantum-chaotic sensors

Lukas J. Fiderer[1] & Daniel Braun[1]

Quantum metrology promises high-precision measurements of classical parameters with far reaching implications for science and technology. So far, research has concentrated almost exclusively on quantum-enhancements in integrable systems, such as precessing spins or harmonic oscillators prepared in non-classical states. Here we show that large benefits can be drawn from rendering integrable quantum sensors chaotic, both in terms of achievable sensitivity as well as robustness to noise, while avoiding the challenge of preparing and protecting large-scale entanglement. We apply the method to spin-precession magnetometry and show in particular that the sensitivity of state-of-the-art magnetometers can be further enhanced by subjecting the spin-precession to non-linear kicks that renders the dynamics chaotic.

[1] Institute for Theoretical Physics, University of Tübingen, Auf der Morgenstelle 14, 72076 Tübingen, Germany. Correspondence and requests for materials should be addressed to L.J.F. (email: lukas.fiderer@uni-tuebingen.de) or to D.B. (email: daniel.braun@uni-tuebingen.de)

Quantum-enhanced measurements (QEM) use quantum effects in order to measure physical quantities with larger precision than what is possible classically with comparable resources. QEMs are therefore expected to have large impact in many areas, such as improvement of frequency standards[1–5], gravitational wave detection[6,7], navigation[8], remote sensing[9], or measurement of very small magnetic fields[10]. A well-known example is the use of so-called NOON states in an interferometer, where a state with $N$ photons in one arm of the interferometer and zero in the other is superposed with the opposite situation[11]. It was shown that the smallest phase shift that such an interferometer could measure scales as $1/N$, a large improvement over the standard $1/\sqrt{N}$ behavior that one obtains from ordinary laser light. The latter scaling is known as the standard quantum limit (SQL), and the $1/N$ scaling as the Heisenberg limit (HL). So far the SQL has been beaten only in few experiments, and only for small $N$ (see e.g.,[3,12,13]), as the required non-classical states are difficult to prepare and stabilize and are prone to decoherence.

Sensing devices used in quantum metrology so far have been based almost exclusively on integrable systems, such as precessing spins (e.g., nuclear spins, NV centers, etc.) or harmonic oscillators (e.g., modes of an electro-magnetic field or mechanical oscillators), prepared in non-classical states (see ref.[14] for a recent review). The idea of the present work is to achieve enhanced measurement precision with readily accessible input states by disrupting the parameter coding by a sequence of controlled pulses that renders the dynamics chaotic. At first sight this may appear a bad idea, as measuring something precisely requires well-defined, reproducible behavior, whereas classical chaos is associated with unpredictible long-term behavior. However, the extreme sensitivity to initial conditions underlying classically chaotic behavior is absent in the quantum world with its unitary dynamics in Hilbert space that preserves distances between states. In turn, quantum-chaotic dynamics can lead to exponential sensitivity with respect to parameters of the system[15].

The sensitivity to changes of a parameter of quantum-chaotic systems has been studied in great detail with the technique of Loschmidt echo[16], which measures the overlap between a state propagated forward with a unitary operator and propagated backward with a slightly perturbed unitary operator. In the limit of infinitesimally small perturbation, the Loschmidt echo turns out to be directly related to the quantum Fisher information (QFI) that determines the smallest uncertainty with which a parameter can be estimated. Hence, a wealth of known results from quantum chaos can be immediately translated to study the ultimate sensitivity of quantum-chaotic sensors. In particular, linear response expressions for fidelity can be directly transfered to the exact expressions for the QFI.

Ideas of replacing entanglement creation by dynamics were proposed previously[17–21], but focussed on initial state preparation, or robustness of the readout[22,23], without introducing or exploiting chaotic dynamics during the parameter encoding. They are hence comparable to spin-squeezing of the input state[24]. Quantum chaos is also favorable for state tomography of random initial states with weak continuous time measurement[25,26], but no attempt was made to use this for precision measurements of a parameter. A recent review of other approaches to quantum-enhanced metrology that avoid initial entanglement can be found in ref.[27].

We study quantum-chaotic enhancement of sensitivity at the example of the measurement of a classical magnetic field with a spin-precession magnetometer. In these devices that count amongst the most sensitive magnetometers currently available[28–32], the magnetic field is coded in a precession frequency of atomic spins that act as the sensor. We show that the precision of the magnetic-field measurement can be substantially enhanced by

non-linearly kicking the spin during the precession phase and driving it into a chaotic regime. The initial state can be chosen as an essentially classical state, in particular a state without initial entanglement. The enhancement is robust with respect to decoherence or dissipation. We demonstrate this by modeling the magnetometer on two different levels: firstly as a kicked top, a well-known system in quantum chaos to which we add dissipation through superradiant damping; and secondly with a detailed realistic model of a spin-exchange-relaxation-free atom-vapor magnetometer including all relevant decoherence mechanisms[28,33], to which we add non-linear kicks.

## Results

**Physical model of a quantum-chaotic sensor.** As a sensor we consider a kicked top (KT), a well-studied quantum-chaotic system[34–36] described by the time-dependent Hamiltonian

$$H_{KT}(t) = \alpha J_z + \frac{k}{2J}J_y^2 \sum_{n=-\infty}^{\infty} \tau\delta(t - n\tau), \qquad (1)$$

where $J_i$ ($i = x, y, z$) are components of the (pseudo-)angular momentum operator, $J \equiv j + 1/2$, and we set $\hbar = 1$. $J_z$ generates a precession of the (pseudo-)angular momentum vector about the $z$-axis with precession angle $\alpha$ which is the parameter we want to estimate. "Pseudo" refers to the fact that the physical system need not be an actual physical spin, but can be any system with $2j + 1$ basis states on which the $J_i$ act accordingly. For a physical spin-$j$ in a magnetic field $B$ in $z$-direction, $\alpha$ is directly proportional to $B$. The $J_y^2$-term is the non-linearity, assumed to act instantaneously compared to the precession, controlled by the kicking strength $k$ and applied periodically with a period $\tau$ that leads to chaotic behavior. The system can be described stroboscopically with discrete time $t$ in units of $\tau$ (set to $\tau = 1$ in the following),

$$|\psi(t)\rangle = U_\alpha(k)|\psi(t-1)\rangle = U_\alpha^t(k)|\psi(0)\rangle \qquad (2)$$

with the unitary Floquet-operator

$$U_\alpha(k) = T\exp\left(-i\int_t^{t+1} dt' H_{KT}(t')\right) = e^{-ik\frac{J_y^2}{2J}}e^{-i\alpha J_z} \qquad (3)$$

that propagates the state of the system from right after a kick to right after the next kick[34–36]. $T$ denotes time-ordering. The total spin is conserved, and $1/J$ can be identified with an effective $\hbar$, such that the limit $j \to \infty$ corresponds to the classical limit, where $X = J_x/J$, $Y = J_y/J$, $Z = J_z/J$ become classical variables confined to the unit sphere. $(Z, \phi)$ can be identified with classical phase space variables, where $\phi$ is the azimuthal angle of $\mathbf{X} = (X, Y, Z)$[36]. For $k = 0$, the dynamics is integrable, as the precession conserves $Z$ and increases $\phi$ by $\alpha$ for each application of $U_\alpha(0)$. Phase space portraits of the corresponding classical map show that for $k \lesssim 2.5$, the dynamics remains close to integrable with large visible Kolmogorov–Arnold–Moser tori, whereas for $k \gtrsim 3.0$ the chaotic dynamics dominates[36].

States that correspond most closely to classical phase space points located at $(\theta, \phi)$ are SU(2)-coherent states ("spin-coherent states", or "coherent states" for short), defined as

$$|j, \theta, \phi\rangle = \sum_{m=-j}^{j} \sqrt{\binom{2j}{j-m}} \sin(\theta/2)^{j-m} \cos(\theta/2)^{j+m} e^{i(j-m)\phi}|jm\rangle \qquad (4)$$

in the usual notation of angular momentum states $|jm\rangle$ (eigenbasis of $\mathbf{J}^2$ and $J_z$ with eigenvalues $j(j+1)$ and $m$, $2j \in \mathbb{N}$, $m = -j, -j+1, \ldots, j$). They are localized at polar and azimuthal angles $\theta, \phi$ with smallest possible uncertainty of all spin-$j$ states

**Fig. 1** Schematic representation of the parameter encoding: propagation starts on the left with an initial state $\rho$ and ends on the right with a measurement (semi-circle symbol). The encoding through linear precession $R_z(\alpha)$ about the z-axis by an angle $\alpha$ is periodically disrupted through parameter independent, non-linear, controlled kicks (blue triangles) that can render the system chaotic

(associated circular area $\sim 1/j$ in phase space). They remain coherent states under the action of $U_\alpha(0)$, i.e., just get rotated, $\phi \mapsto \phi + \alpha$. For the KT, the parameter encoding of $\alpha$ in the quantum state breaks with the standard encoding scheme (initial state preparation, parameter-dependent precession, measurement) by periodically disrupting the coding evolution with parameter-independent kicks that generate chaotic behavior (see Fig. 1).

An experimental realization of the kicked top was proposed in ref. [37], including superradiant dissipation. It has been realized experimentally[38] in cold cesium vapor using optical pulses (see Supplementary Note 1 for details).

**Quantum parameter estimation theory.** Quantum measurements are most conveniently described by a positive-operator valued measure (POVM) $\{\Pi_\xi\}$ with positive operators $\Pi_\xi$ (POVM elements) that fulfill $\int d\xi \Pi_\xi = 1$. Measuring a quantum state described by a density operator $\rho_\alpha$ yields for a given POVM and a given parameter $\alpha$ encoded in the quantum state a probability distribution $p_\alpha(\xi) = \text{tr}(\Pi_\xi \rho_\alpha)$ of measurement results $\xi$. The Fisher information $I_{\text{Fisher},\alpha}$ is then defined by

$$I_{\text{Fisher},\alpha} := \int d\xi \frac{(dp_\alpha(\xi)/d\alpha)^2}{p_\alpha(\xi)}. \quad (5)$$

The minimal achievable uncertainty, i.e., the variance of the estimator $\text{Var}(\alpha_{\text{est}})$, with which a parameter $\alpha$ of a state $\rho_\alpha$ can be estimated for a given POVM with $M$ independent measurements is given by the Cramér–Rao bound, $\text{Var}(\alpha_{\text{est}}) \geq 1/(M I_{\text{Fisher},\alpha})$. Further optimization over all possible (POVM-)measurements leads to the quantum-Cramér–Rao bound (QCRB),

$$\text{Var}(\alpha_{\text{est}}) \geq \frac{1}{M I_\alpha}, \quad (6)$$

which presents an ultimate bound on the minimal achievable uncertainty, where $I_\alpha$ is the quantum Fisher information (QFI), and $M$ the number of independent measurements[39].

The QFI is related to the Bures distance $ds_{\text{Bures}}^2$ between the states $\rho_\alpha$ and $\rho_{\alpha+d\alpha}$, separated by an infinitesimal change of the parameter $\alpha$, $ds_{\text{Bures}}^2(\rho, \sigma) \equiv 2(1 - \sqrt{F(\rho, \sigma)})$. The fidelity $F(\rho, \sigma)$ is defined as $F(\rho, \sigma) = \left\| \rho^{1/2} \sigma^{1/2} \right\|_1^2$, and $\|A\|_1 \equiv \text{tr}\sqrt{AA^\dagger}$ denotes the trace norm[40]. With this[41],

$$I_\alpha = 4 ds_{\text{Bures}}^2(\rho_\alpha, \rho_{\alpha+d\alpha})/d\alpha^2. \quad (7)$$

For pure states $\rho = |\psi\rangle\langle\psi|$, $\sigma = |\phi\rangle\langle\phi|$, the fidelity is simply given by $F(\rho, \sigma) = |\langle\psi|\phi\rangle|^2$. A parameter coded in a pure state via the unitary transformation $|\psi_\alpha\rangle = e^{-i\alpha G}|\psi(0)\rangle$ with hermitian generator $G$ gives the QFI[42]

$$I_\alpha = 4 \text{Var}(G) \equiv 4(\langle G^2 \rangle - \langle G \rangle^2), \quad (8)$$

which holds for all $\alpha$, and where $\langle \cdot \rangle \equiv \langle\psi_\alpha| \cdot |\psi_\alpha\rangle$.

**Loschmidt echo.** The sensitivity to changes of a parameter of quantum-chaotic systems has been studied in great detail with the technique of Loschmidt echo[16], which measures the overlap $F_\epsilon(t)$ between a state propagated forward with a unitary operator $U_\alpha(t)$ and propagated backward with a slightly perturbed unitary operator $U_{\alpha+\epsilon}(-t) = U_{\alpha+\epsilon}^\dagger(t)$, where $U_\alpha(t) = T\exp\left(-\frac{i}{\hbar}\int_0^t dt' H_\alpha(t')\right)$ with the time ordering operator $T$, the Hamiltonian, $H_{\alpha+\epsilon}(t) = H_\alpha(t) + \epsilon V(t)$ and the perturbation $V(t)$,

$$F_\epsilon(t) = |\langle\psi(0)|U_\alpha(t)U_{\alpha+\epsilon}(-t)\psi(0)\rangle|^2. \quad (9)$$

$F_\epsilon$ is exactly the fidelity that enters via the Bures distance in the definition Eq. (7) of the QFI for pure states, such that $I_\alpha(t) = \lim_{\epsilon \to 0} 4 \frac{1 - F_\epsilon(t)}{\epsilon^2}$.

**Benchmarks.** In order to assess the influence of the kicking on the QFI, we calculate as benchmarks the QFI for the (integrable) top with Floquet operator $U_\alpha(0)$ without kicking, both for an initial coherent state and for a Greenberger–Horne–Zeilinger (GHZ) state $|\psi_{\text{GHZ}}\rangle = (|j, j\rangle + |j, -j\rangle)/\sqrt{2}$. The latter is the equivalent of a NOON state written in terms of (pseudo-)angular momentum states. The QFI for the time evolution Eq. (2) of a top with Floquet operator $U_\alpha(0)$ is given by Eq. (8) with $G = J_z$. For an initial coherent state located at $\theta, \phi$ it results in a QFI

$$I_\alpha(t) = 2t^2 j \sin^2 \theta. \quad (10)$$

As expected, $I_\alpha(t) = 0$ for $\theta = 0$ where the coherent state is an eigenstate of $U_\alpha(0)$. The scaling $\propto t^2$ is typical of quantum coherence, and $I_\alpha(t) \propto j$ signifies a SQL-type scaling with $N = 2j$, when the spin-$j$ is composed of $N$ spin-$\frac{1}{2}$ particles in a state invariant under permutations of particles. For the benchmark, we use the optimal value $\theta = \pi/2$ in Eq. (10), i.e., $I_{\text{top,CS}} \equiv 2t^2 j$. For a GHZ state, the QFI becomes

$$I_\alpha(t) = 4t^2 j^2 \equiv I_{\text{top,GHZ}}, \quad (11)$$

which clearly displays the HL-type scaling $\propto (2j)^2 \equiv N^2$.

**Results for the kicked top without dissipation.** In the fully chaotic case, known results for the Loschmidt echo suggest a QFI of the KT $\propto t j^2$ for times $t$ with $t_E < t < t_H$, where $t_E = \frac{1}{\lambda}\ln\left(\frac{\Omega_V}{h^d}\right)$ is the Ehrenfest time, and $t_H = \hbar/\Delta$ the Heisenberg time; $\lambda$ is the Lyapunov exponent, $\Omega_V$ the volume of phase-space, $h^d$ with $d$ the number of degrees of freedom the volume of a Planck cell, and $\Delta$ the mean energy level spacing[16,36,43]. For the kicked top, $h^d \simeq \Omega_V/(2J)$. More precisely, we find for $t \simeq t_E$ a QFI $I_\alpha \propto t j^2$ and for $t \gg t_H$ (see Methods)

$$I(t) = 8 s \sigma_{\text{cl}} t^2 J, \quad (12)$$

where $s$ denotes the number of invariant subspaces $s$ of the Hilbert space ($s = 3$ for the kicked top with $\alpha = \pi/2$, see page 359 in ref. [15]), and $\sigma_{\text{cl}}$ is a transport coefficient that can be calculated numerically. The infinitesimally small perturbation relevant for the QFI makes that one is always in the perturbative regime[44,45]. The Gaussian decay of Loschmidt echo characteristic of that regime becomes the slower the smaller the perturbation and goes over into a power law in the limit of infinitesimally small perturbation[16].

The numerical results for the QFI in Fig. 2 illustrate a crossover of power-law scalings in the fully chaotic case ($k = 30$) for an initial coherent state located on the equator $(\theta, \phi) = (\pi/2, \pi/2)$. The analytical Loschmidt echo results are nicely reproduced: a smooth transition in scaling from $t j^2 \to t^2 j$ for $t = t_E \to t \gtrsim t_H$ can be observed and confirms Eqs. (15) and (16) in the Methods for

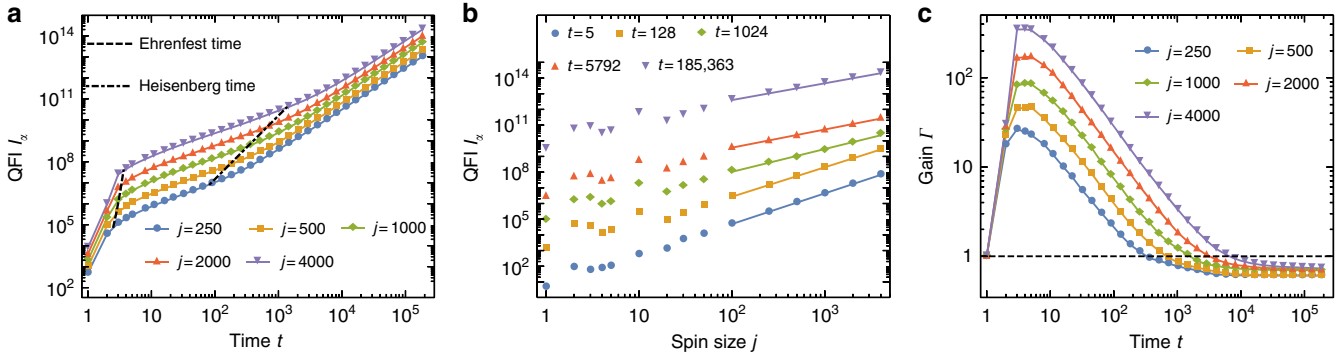

**Fig. 2** Quantum-chaotic enhancement of sensitivity. **a** $t$-scaling of the quantum Fisher information (QFI) $I_\alpha$ in the strongly chaotic case. Dashed and dash-dotted lines indicate Ehrenfest and Heisenberg times, respectively. **b** $j$-scaling of the QFI. Fits have slopes 1.96, 1.88, 1.46, 1.16, and 1.08 in increasing order of $t$. **c** Gain $\Gamma = I_{\alpha,\mathrm{KT}}/I_{\mathrm{top,CS}}$ as function of time $t$ for different values of spin size $j$. The dashed black line marks the threshold $\Gamma = 1$. Kicking strength $k = 30$, and initial coherent state at $(\theta, \phi) = (\pi/2, \pi/2)$ in all plots

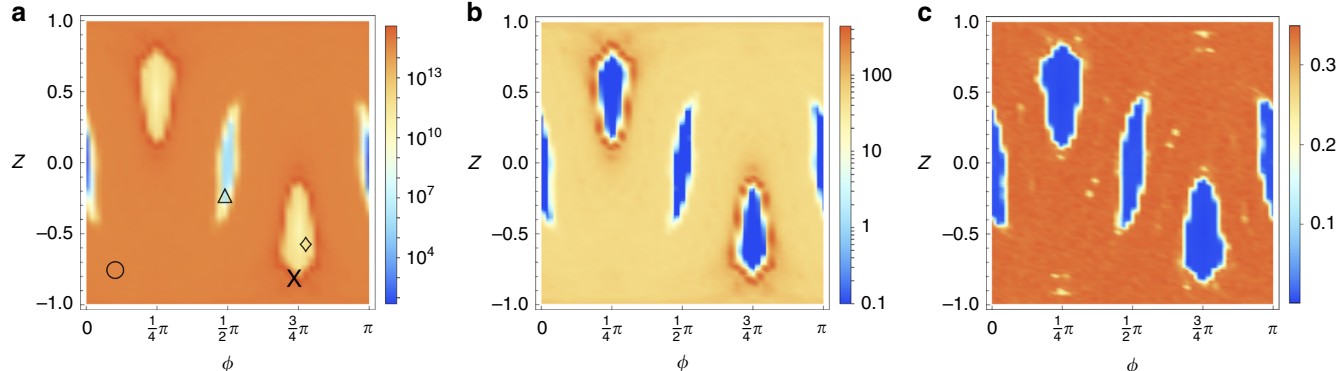

**Fig. 3** Portraits of a mixed phase space for large spin size $j = 4000$. The heat maps depict quantum Fisher information (QFI) in panel **a** and gain $\Gamma$ in panel **b** at time $t = 2^{15}$ and kicking strength $k = 3$. $\Gamma < 0.1$ is depicted as $\Gamma = 0.1$ to simplify the representation. A gain of more than two orders of magnitude in the QFI is observed for edge states, localized on the border of the stability islands. For comparison, panel **c** depicts the classical Lyapunov exponent $\lambda$

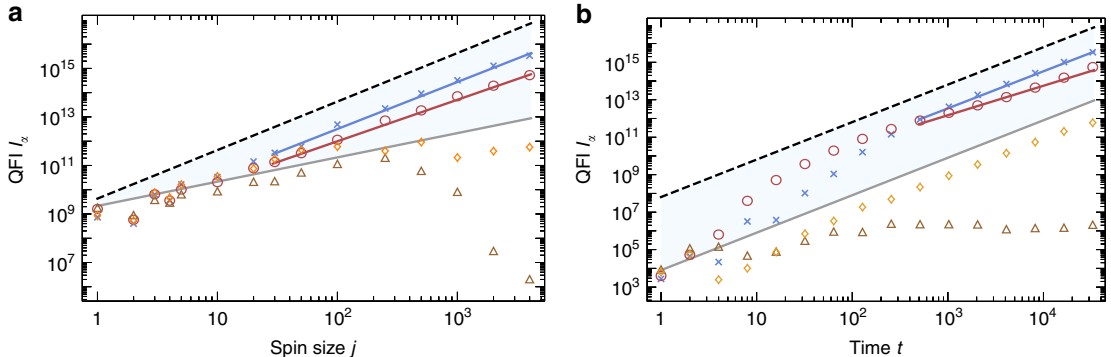

**Fig. 4** Scaling of quantum Fisher information (QFI) in different areas of a mixed phase space. QFI $I_\alpha(j)$ (panel **a** for time $t = 2^{15}$) and $I_\alpha(t)$ (panel **b** for spin size $j = 4000$) for kicking strength $k = 3$ and different initial coherent states $|\theta, \phi\rangle$ (as marked in panel **a** of Fig. 3): inside an equatorial (brown triangles, $|1.82, 1.54\rangle$) and a non-equatorial (orange diamonds, $|2.20, 2.44\rangle$) stability island, in the chaotic sea (red circles, $|2.46, 0.32\rangle$), and an edge state (blue crosses, $|2.56, 2.31\rangle$). Benchmarks $I_{\mathrm{top,CS}}$ (gray line) and $I_{\mathrm{top,GHZ}}$ (black dashed line) represent the standard quantum limit and Heisenberg limit, respectively. Fits for the edge state and the state in the chaotic sea exhibit a slope of 1.94 and 1.71 for the $j$-scaling, and 1.98 and 1.57 for the $t$-scaling

$t > t_\mathrm{E} = \ln(2J)/\lambda$, with the numerically determined Lyapunov exponent $\lambda \simeq 2.4733$, and Eq. (12) for $t \gtrsim t_\mathrm{H} \simeq J/3^{16}$. We find for relatively large $j$ ($j \gtrsim 10^2$) a scaling $I_\alpha \propto j^{1.08}$ in good agreement with Eq. (12) predicting a linear $j$-dependence for large $t \gtrsim t_\mathrm{H}$. During the transient time $t < t_\mathrm{E}$, when the state is spread over the phase space, QFI shows a rapid growth that can be attributed to the generation of coherences that are particularly sensitive to the precession.

The comparison of the KT's QFI $I_{\alpha,\mathrm{KT}}$ with the benchmark $I_{\mathrm{top,CS}}$ of the integrable top in Fig. 2c shows that a gain of more than two orders of magnitude for $j = 4000$ can be found at $t \lesssim t_\mathrm{E}$. Around $t_\mathrm{E}$ the state has spread over the phase space and has developed coherences while for larger times $t > t_\mathrm{E}$ the top catches up due to its superior time scaling ($t^2$ vs. $t$). The long-time behavior yields a constant gain <1, which means that the top

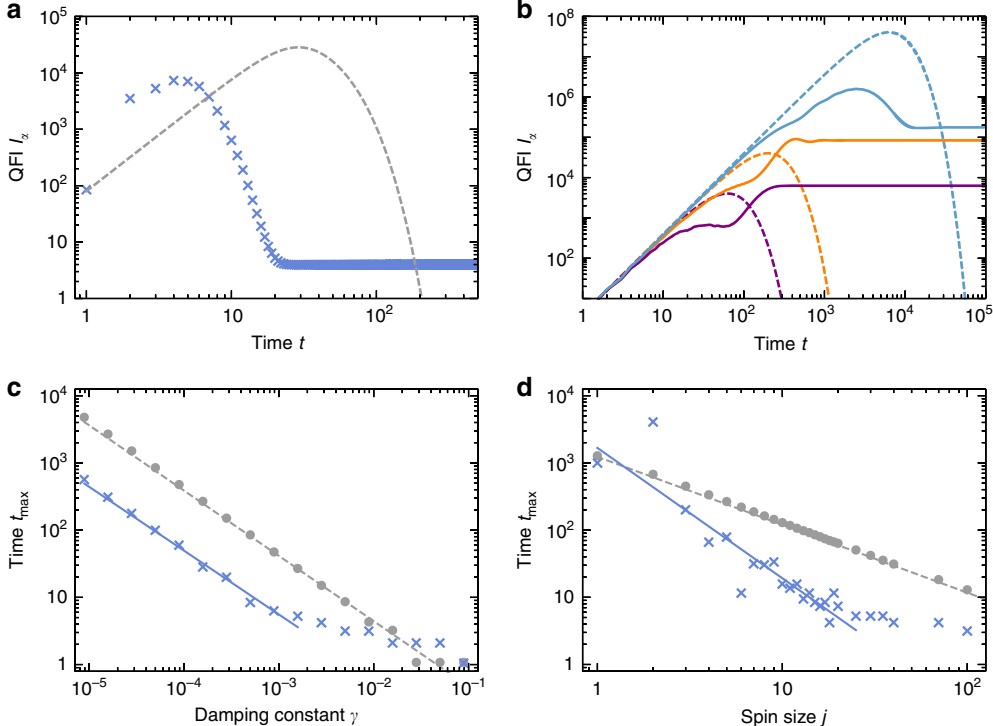

**Fig. 5** Time evolution of the quantum Fisher information (QFI) for the dissipative kicked top. In all subplots, we look at an initial coherent state at $(\theta, \phi) = (\pi/2, \pi/2)$. **a** Exemplary behavior as function of time for a dissipative top (DT, dashed gray line) and a chaotic (kicking strength $k = 30$) dissipative kicked top (DKT, blue crosses) for spin size $j = 40$, damping constant $\gamma = 0.5 \times 10^{-3}$. **b** High plateau values of the QFI are achieved for $j = 2$ and $k = 30$ for various values of the damping constant ($\gamma = 0.5 \times 10^{-2}$ for the purple line, $\gamma = 1.58 \times 10^{-3}$ for the orange line, and $\gamma = 0.5 \times 10^{-4}$ for the blue line), while dashed lines represent corresponding QFI with $k = 0$. **c, d** Time $t_{\max}$ at which the QFI of the DT (gray dots) and DKT (blue crosses) reaches its maximum as function of $\gamma$ and $j$, respectively. The fits for the DT (gray line) exhibit a slope of −0.97 and −1.01, and the fits for the DKT (blue, dashed) a slope of −0.95 and −1.94 for scaling with $\gamma$ and $j$, respectively. Parameters $j = 15$ in **c** and $\gamma = 0.5 \times 10^{-3}$ in **d**. The plotted range of the fits corresponds to the fitted data

achieves a higher QFI than the KT in this regime. The gain becomes constant as both top and KT exhibit a $t^2$ scaling of the QFI.

Whereas in the fully chaotic regime the memory of the initial state is rapidly forgotten, and the initial state can therefore be chosen anywhere in the chaotic sea without changing much the QFI, the situation is very different in the case of a mixed phase space, in which stability islands are still present. Figure 3 shows phase space distributions for $k = 3$ of QFI and gain $\Gamma = I_{\alpha,KT}/I_{top,CS}$ exemplarily for large $t$ and $j$ ($t = 2^{15}$, $j = 4000$) in comparison with the classical Lyapunov exponent, where $\phi$, $Z$ signify the position of the initial coherent state.

The QFI nicely reproduces the essential structure in classical phase space for the Lyapunov exponent $\lambda$ (see Methods for the calculation of $\lambda$). Outside the regions of classically regular motion, the KT clearly outperforms the top by more than two orders of magnitude. Remarkably, the QFI is highest at the boundary of non-equatorial islands of classically regular motion. Coherent states located on that boundary will be called edge states.

The diverse dynamics for different phase space regions calls for dedicated analyses. Figure 4 depicts the QFI with respect to $j$ and $t$. The blue area is lower and upper bounded by the benchmarks $I_{top,CS}$ and $I_{top,GHZ}$, Eq. (11), respectively.

We find that initial states in the chaotic sea perform best for small times while for larger times ($t \gtrsim 300$ for $j = 4000$) edge states perform best. Note that this is numerically confirmed up to very high QFI values ($>10^{14}$). The superiority of edge states holds for $j \gtrsim 10$ for large times ($t \gtrsim 10^3$, see Fig. 4a). Large values of $j$ allow one to localize states essentially within a stability island. A coherent state localized within a non-equatorial stability island shows a quadratic $t$-scaling analog to the regular top. For a state

$|\psi_{eq}\rangle$ localized around a point within an equatorial island of stability the QFI drastically decays with increasing $j$ (brown triangles). The scaling with $t$ for $j = 4000$ reveals that QFI does not increase with $t$ in this case, it freezes. One can understand the phenomenon as arising from a freeze of fidelity due to a vanishing time averaged perturbation[16,46]: the dynamics restricts the states to the equatorial stability island with time average $\overline{\langle \psi_{eq}|(U_\alpha^t)^\dagger(k)J_z U_\alpha^t(k)|\psi_{eq}\rangle} = 0$. This can be verified numerically, and contrasted with the dynamics when initial states are localized in the chaotic sea or on a non-equatorial island.

**Results for the dissipative kicked top**. For any quantum-enhanced measurement, it is important to assess the influence of dissipation and decoherence. We first study superradiant damping[47–51] as this enables a proof-of-principle demonstration with an analytically accessible propagator for the master equation with spins up to $j \simeq 200$ and correspondingly large gains. Then, in the next subsection, we show by detailed and realistic modeling including all the relevant decoherence mechanisms that sensitivity of existing state-of-the-art alkali-vapor-based spin-precession magnetometers in the spin-exchange-relaxation-free (SERF) regime can be enhanced by non-linear kicks.

At sufficiently low temperatures ($k_B T \ll \hbar\omega$, where $\hbar\omega$ is the level spacing between adjacent states $|jm\rangle$) superradiance is described by the Markovian master equation for the spin-density matrix $\rho(t)$ with continuous time,

$$\frac{d}{dt}\rho(t) = \gamma([J_-, \rho(t)J_+] + [J_-\rho(t), J_+]) \equiv \rho(t), \qquad (13)$$

where $J_\pm \equiv J_x \pm iJ_y$, with the commutator $[A, B] = AB - BA$, and $\gamma$ is the dissipation rate, with the formal solution $\rho(t) = \exp(\Lambda t)\rho(0) \equiv D(t)\rho(0)$. The full evolution is governed by $d\rho(t)/dt = \Lambda\rho(t) - i\hbar[H_{KT}(t), \rho(t)]$. Dissipation and precession about the $z$-axis commute, $\Lambda(J_z \rho J_z) = J_z(\Lambda\rho)J_z$. $\Lambda$ can therefore act permanently, leading to the propagator $P$ of $\rho$ from discrete time $t$ to $t + \tau$ for the dissipative kicked top (DKT)[36,52]

$$\rho(t + \tau) = P\rho(t) = U_\alpha(k)(D(\tau)\rho(t))U_\alpha^\dagger(k). \tag{14}$$

For the sake of simplicity, we again set the period $\tau = 1$, and $t$ is taken again as discrete time in units of $\tau$. Then, $\gamma\tau \equiv \gamma$ controls the effective dissipation between two unitary propagations. Classically, the DKT shows a strange attractor in phase space with a fractal dimension that reduces from $d = 2$ at $\gamma = 0$ to $d = 0$ for large $\gamma$, when the attractor shrinks to a point attractor and migrates towards the ground state $|j, -j\rangle$[52]. Quantum mechanically, one finds a Wigner function with support on a smeared out version of the strange attractor that describes a non-equilibrium steady state reached after many iterations. Such a non-trivial state is only possible through the periodic addition of energy due to the kicking. Because of the filigrane structure of the strange attractor, one might hope for relatively large QFI, whereas without kicking the system would decay to the ground state, where the QFI vanishes. Creation of steady non-equilibrium states may therefore offer a way out of the decoherence problem in quantum metrology, see also section V.C in ref.[27] for similar ideas.

Vanishing kicking strength, i.e., the dissipative top (DT) obtained from the DKT by setting $k = 0$, will serve again as benchmark. While in the dissipation-free regime, we took the top's QFI and with it its SQL-scaling ($\propto jt^2$) as reference, SQL-scaling no longer represents a proper benchmark, because damping typically corrupts QFI with increasing time. To illustrate the typical behavior of QFI, we exemplarily choose certain spin sizes $j$ and damping constants $\gamma$ here and in the following, such as $j = 40$ and $\gamma = 0.5 \times 10^{-3}$ in Fig. 5a, while computational limitations restrict us to $j \lesssim 200$.

Figure 5 shows the typical overall behavior of the QFI of the DT and DKT as function of time: after a steep initial rise $\propto t^2$, the QFI reaches a maximum whose value is the larger the smaller the dissipation. Then the QFI decays again, dropping to zero for the DT, and a plateau value for the DKT. The time at which the maximum value is reached decays roughly as $1/(j\gamma)$ for the DT, and as $1/j^{0.95}$ and $1/\gamma^{1.94}$ for the DKT. The plateau itself is in general relatively small for the limited values of $j$ that could be investigated numerically, but it should be kept in mind that (i) for the DT the plateau does not even exist (QFI always decays to zero for large time, as dissipation drives the system to the ground state $|j, -j\rangle$ which is an eigenstate of $J_z$ and hence insensitive to precession); and (ii) there are exceptionally large plateau values even for small $j$, see e.g., the case of $j = 2$ in Fig. 5b. There, for $\gamma = 1.58 \times 10^{-3}$, the plateau value is larger by a factor 2.35 than the DT's QFI optimized over all initial coherent states for all times. Note that since $\Lambda(J_z \rho J_z) = J_z(\Lambda\rho)J_z$, for the DT an initial precession about the $z$-axis that is part of the state preparation can be moved to the end of the evolution and does not influence the QFI of the DT. Optimizing over the initial coherent state can thus be restricted to optimizing over $\theta$.

When considering dynamics, it is natural also to include time as a resource. Indeed, experimental sensitivities are normally given as uncertainties per square root of Hertz: longer (classical) averaging reduces the uncertainty as $1/\sqrt{T_{av}}$ with averaging time $t = T_{av}$. For fair comparisons, one multiplies the achieved uncertainty with $\sqrt{T_{av}}$. Correspondingly, we now compare rescaled QFI and Fisher information, namely $I_\alpha^{(t)} \equiv I_\alpha/T_{av}$,

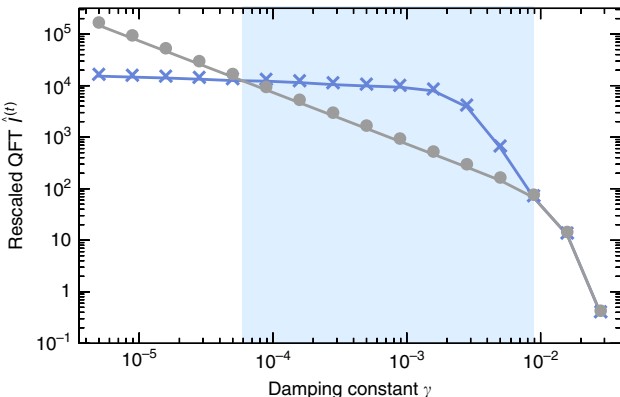

**Fig. 6** Enhancement in measurement precision through kicking (blue crosses) is found over a broad range of damping strengths. Comparison of the maximal rescaled quantum Fisher information $\hat{I}_\alpha^{(t)}$ for the dissipative top (gray dots) and the dissipative kicked top (blue crosses); kicking strength $k = 30$ and spin size $j = 100$. In both cases, $\hat{I}_\alpha^{(t)}$ was optimized over the location of initial coherent states. The blue-shaded area marks the range $1.2 \lesssim 2\gamma j^2 \lesssim 180$ where the reference is outperformed

$I_{Fisher,\alpha}^{(t)} \equiv I_{Fisher,\alpha}/T_{av}$. A protocol that reaches a given level of QFI more rapidly has then an advantage, and best precision corresponds to the maximum rescaled QFI or Fisher information, $\hat{I}_\alpha^{(t)} \equiv \max_t I_\alpha^{(t)}$ or $\hat{I}_{Fisher,\alpha}^{(t)} \equiv \max_t I_{Fisher,\alpha}^{(t)}$.

Figure 6 shows that in a broad range of dampings that are sufficiently strong for the QFI to decay early, the maximum rescaled QFI of the DKT beats that quantity of the DT by up to an order of magnitude. Both quantities were optimized over the location of the initial coherent states.

Figure 7 shows $\hat{I}_\alpha^{(t)}$ and the gain in that quantity compared to the non-kicked case as function of both the damping and the kicking strength. One sees that in the intermediate damping regime ($\gamma \simeq 10^{-3}$) the gain increases with kicking strength, i.e., increasingly chaotic dynamics.

For exploiting the enhanced sensitivity shown to exist through the large QFI, one needs also to specify the actual measurement of the probe. In principle, the QCRB formalism allows one to identify the optimal POVM measurement if the parameter is known, but these may not always be realistic. In Fig. 8, we investigate $J_y$ as a feasible example for a measurement for a spin size $j = 200$ after $t = 2$ time steps. We find that there exists a broad range of kicking strengths where the reference (state-optimized but $k = 0$) is outperformed in both cases, with and without dissipation. In a realistic experiment, control parameters such as the kicking strength are subjected to variations. A 5% variance in $k$, which was reported in ref.[38], reduces the Fisher information only marginally and does not challenge the advantage of kicking. This can be calculated by rewriting the probability that enters in the Fisher information in Eq. (5) according to the law of total probability, $p_\alpha(\xi) = \int dk\, p(k) p_\alpha(\xi|k)$ where $p(k)$ is an assumed Gaussian distribution of $k$ values with 5% variance and $p_\alpha(\xi|k) = \text{tr}[\Pi_\xi \rho_\alpha(k)]$ with $\Pi_\xi$ a POVM element and $\rho_\alpha(k)$ the state for a given $k$ value. The advantage from kicking remains when investigating a rescaled and time-optimized Fisher information (not shown in Fig. 8).

**Improving a SERF magnetometer.** We finally show that quantum-chaotically enhanced sensitivity can be achieved in state-of-the-art magnetometers by investigating a rather realistic and detailed model of an alkali-vapor-based spin-precession magnetometer acting in the SERF regime. SERF magentometers count amongst the most sensitive magnetometers for detecting

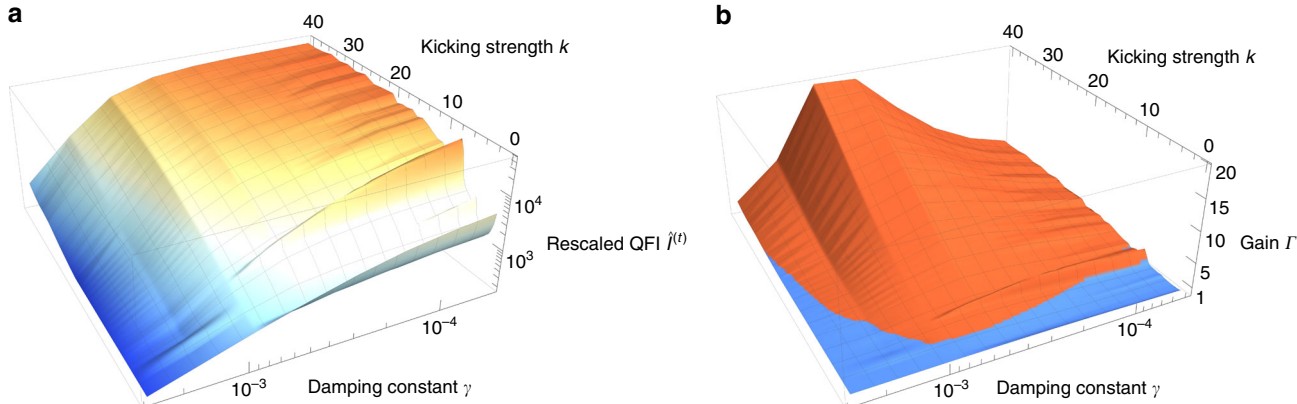

**Fig. 7** Rescaled quantum Fisher information (QFI) and gains in measurement precision. Rescaled QFI of the dissipative kicked top $\hat{l}^{(t)}_{\alpha,\mathrm{DKT}}(\theta = \pi/2, \phi = \pi/2)$ in panel **a** and gain $\Gamma = \frac{\hat{l}^{(t)}_{\alpha,\mathrm{DKT}}(\theta=\pi/2,\phi=\pi/2)}{\hat{l}^{(t)}_{\alpha,\mathrm{DT}}}$ in panel **b** as function of damping constant $\gamma$ and kicking strength $k$ at $j = 200$. Note that in this case $\hat{l}^{(t)}_{\alpha}$ is optimized over initial states only for the dissipative top

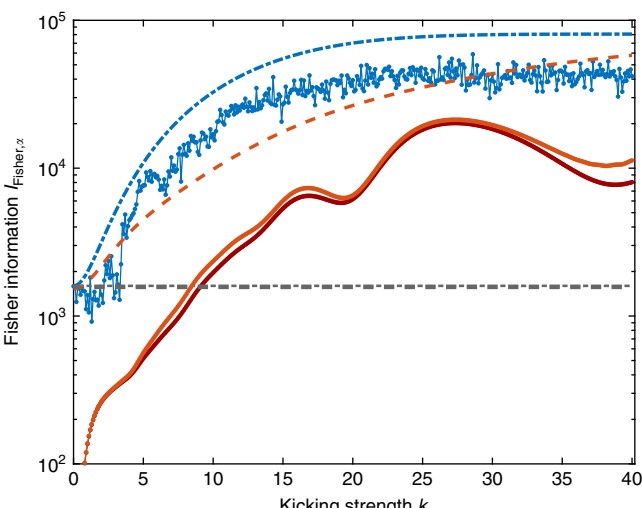

**Fig. 8** Performance of an exemplary spin-component measurement with superradiance damping. Fisher information $I_{\mathrm{Fisher},\alpha}$ related to measuring the $y$-component of the spin, $J_y$, after $t = 2$ time steps without dissipation (blue line) as well as in the presence of dissipation (bright red line, damping constant $\gamma = 0.5 \times 10^{-3}$), upper bounded by corresponding quantum Fisher informations (blue dash-dotted and red dashed lines), for spin size $j = 200$, and an initial state at $(\theta, \phi) = (\pi/2, \pi/2)$. For the dissipative case, Fisher information with a 5% uncertainty in the kicking strength is given by the dark red line slightly below the bright red line. Horizontal gray lines represent benchmarks (kicking strength $k = 0$) for $t = 2$ optimized over initial states (dashed with dissipation and dash-dotted without; lines are almost on top of each other). In both cases the benchmark is clearly outperformed for $k \gtrsim 3.5$ and $k \gtrsim 10.6$ without and with dissipation, respectively

small quasi-static magnetic fields[28–32]. We consider a cesium-vapor magnetometer at room temperature in the SERF regime similar to experiments with rubidium in ref. [53]. Kicks on the single cesium-atom spins can be realized as in ref. [38] by exploiting the spin-dependent rank-2 (ac Stark) light-shift generated with the help of an off-resonant laser pulse. Typical SERF magnetometers working at higher temperatures with high buffer-gas pressures exhibit an unresolved excited state hyperfine splitting due to pressure broadening, which makes kicks based on rank-2

light-shifts ineffective. Dynamics are modeled in the electronic ground state $6^2S_{1/2}$ of $^{133}$Cs that splits into total spins of $f = 3$ and $f = 4$, where kicks predominantly act on the $f = 3$ manifold.

The model is quite different from the foregoing superradiance model because of a different decoherence mechanism originating from collisions of Cs atoms in the vapor cell: We include spin-exchange and spin-destruction relaxation, as well as additional decoherence induced by the optical implementation of the kicks. With this implementation of kicks one is confined to a small spin size $f = 3$ of single atoms, such that the large improvements in sensitivity found for the large spins discussed above cannot be expected. Nevertheless, we still find a clear gain in the sensitivity and an improved robustness to decoherence due to kicking. Details of the model described with a master equation[33,54] can be found in the Supplementary Note 2.

Spins of cesium atoms are initially pumped into a state spin-polarized in $z$-direction orthogonal to the magnetic field $\mathbf{B} = B\hat{\mathbf{y}}$ in $y$-direction, whose strength $B$ is the parameter $\alpha$ to be measured. We let spins precess in the magnetic field, and, by incorporating small kicks about the $x$-axis, we find an improvement over the reference (without kicks) in terms of rescaled QFI and the precision based on the measurement of the electron-spin component $S_z$ orthogonal to the magnetic field. The best possible measurement precision $\Delta B$ in units of $\mathrm{T}/\sqrt{\mathrm{Hz}}$ per 1 cm$^3$ vapor volume is $\Delta B = 1/\sqrt{nI_B^{(t)}}$ where $n \simeq 2 \times 10^{10}$ is the number of cesium atoms in 1 cm$^3$. For a specific measurement, $I_B^{(t)}$ must be replaced by the corresponding rescaled Fisher information $I_{\mathrm{Fisher},B}^{(t)}$. We compare the models with and without kicks directly on the basis of the Fisher information rather than modeling in addition the specific optical implementation and the corresponding noise of the measurement of $S_z$. Neglecting this additional read-out-specific noise leads to slightly better precision bounds than given in the literature, but does not distort the comparison.

The magnetic field was set to $B = 4 \times 10^{-14}$ T in $y$-direction, such that the condition for the SERF regime is fulfilled, i.e., the Larmor frequency is much smaller than the spin-exchange rate, and the period is set to $\tau = 1$ ms. Since kicks induce decoherence in the atomic spin system, we have to choose a very small effective kicking strength of $k \simeq 6.5 \times 10^{-4}$ for the kicks around the $x$-axis (with respect to the $f = 3$ ground-state manifold), generated with an off-resonant 2 μs light pulse with intensity $I_{\mathrm{kick}} = 0.1$ mW/cm$^2$ linearly polarized in $x$-direction, to find an advantage over the reference.

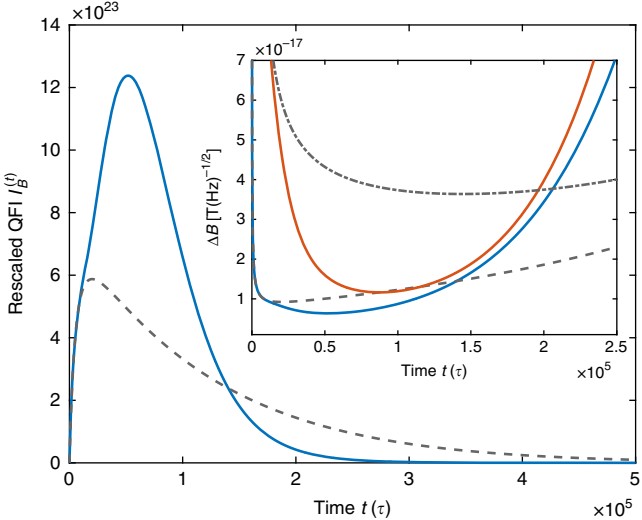

**Fig. 9** Performance of a magnetic-field measurement with a kicked atomic-vapor magnetometer. While the gray dashed line shows the rescaled quantum Fisher information $I_B^{(t)}$ for measuring the magnetic field $B$ with an spin-exchange-relaxation-free magnetometer, the blue line is obtained by adding a short optical kick at the end of each period $\tau$. The inset shows precision $\Delta B$ in units of T per $\sqrt{Hz}$ for an optimal measurement (gray dashed line and blue line without and with kicks, respectively) and for measuring the $z$-component of the electronic spin $S_z$ (gray dash-dotted and red line without and with kicks, respectively)

The example of Fig. 9 shows about 31% improvement in measurement precision $\Delta B$ for an optimal measurement (QFI, upper right inset) and 68% improvement in a comparison of $S_z$ measurements (inset), which is impressive in view of the small system size. The achievable measurement precision of the kicked dynamics exhibits an improved robostness to decoherence: rescaled QFI for the kicked dynamics continues to increase and sets itself apart from the reference around the coherence time associated with spin-destruction relaxation. The laser light for these pulses can be provided by the laser used for the read out, which is typically performed with an off-resonant laser. A further improvement in precision is expected from additionally measuring kick pulses for readout or by applying kicks not only to the $f = 3$ but also to the $f = 4$ ground-state manifold of $^{133}$Cs. Further, it might be possible to dramatically increase the relevant spin-size by applying the kicks to the joint spin of the cesium atoms, for instance, through a double-pass Faraday effect[55].

## Discussion

Rendering the dynamics of quantum sensors chaotic allows one to harvest a quantum enhancement for quantum metrology without having to rely on the preparation or stabilization of highly entangled states. Our results imply that existing magnetic field sensors[31,56] based on the precession of a spin can be rendered more sensitive by disrupting the time-evolution by non-linear kicks. The enhancement persists in rather broad parameter regimes even when including the effects of dissipation and decoherence. Besides a thorough investigation of superradiance damping over large ranges of parameters, we studied a cesium-vapor-based atomic magnetomter in the SERF regime based on a detailed and realistic model[28–31,53]. Although the implementation of the non-linearity via a rank-2 light shift introduces additional decoherence and despite the rather small atomic spin size $f \leq 4$, a considerable improvement in measurement sensitivity is found (68% for a read-out scheme based on the measurement of the electronic spin-component $S_z$). The required non-linearity that

can be modulated as function of time has been demonstrated experimentally in ref. [38] in cold cesium vapor.

Even higher gains in sensitivity are to be expected if an effective interaction can be created between the atoms, as this opens access to larger values of total spin size for the kicks. This may be achieved e.g., via a cavity as suggested for pseudo-spins in ref. [57], or the interaction with a propagating light field as demonstrated experimentally in refs. [55,58] with about $10^{12}$ cesium atoms. More generally, our scheme will profit from the accumulated knowledge of spin-squeezing, which is also based on the creation of an effective interaction between atoms. Finally, we expect that improved precision can be found in other quantum sensors that can be rendered chaotic as well, as the underlying sensitivity to change of parameters is a basic property of quantum-chaotic systems.

## Methods

**QFI for kicked time-evolution of a pure state**. The QFI in the chaotic regime with large system dimension $2J$ and times larger than the Ehrenfest time, $t > t_E$, is given in linear-response theory by an auto-correlation function $C(t) \equiv \langle \bar{V}(t)\bar{V}(0) \rangle - \langle \bar{V}(t) \rangle \langle \bar{V}(0) \rangle$ of the perturbation of the Hamiltonian in the interaction picture, $H_{\alpha+\epsilon}(t) = H_\alpha(t) + \epsilon V(t)$, $\bar{V}(t) = U_\alpha(-t)V(t)U_\alpha(t)$:

$$I_\alpha(t) = 4\left( tC(0) + 2\sum_{t'=0}^{t-1} (t - t')C(t') \right). \quad (15)$$

In our case, the perturbation $V(t) = J_z$ is proportional to the parameter-encoding precession Hamiltonian, and the first summand in Eq. (15) can be calculated for an initial coherent state,

$$C(0) = \frac{1}{3}j(j+1), \quad (16)$$

giving a $tj^2$-scaling starting from $t_E$. Due to the finite Hilbert-space dimension of the kicked top, the auto-correlation function decays for large times to a finite value $\bar{C}$, leading to a term quadratic in $t$ from the sum in Eq. (15) that simplifies to $I_\alpha = 4\bar{C}t^2$ for $t \gg t_H$. If one rescales $J_z \to J_z/J$ such that it has a well defined classical limit, random matrix theory allows us to estimate the average value of $C(t)$ for large times: $\bar{C} = 2Js\sigma_{cl}$, and $\sigma_{cl}$ is a transport coefficient that can be calculated numerically[16]. This yields Eq. (12).

**Lyapunov exponent**. A data point located at $(Z, \phi)$ for the Lyapunov exponent in Fig. 2c was obtained numerically by averaging over 100 initial conditions equally distributed within a circular area of size $1/j$ (corresponding to the coherent state) centered around $(Z, \phi)$.

**Data availability**. Numerical simulation data from this work have been submitted to figshare.com with DOI 10.6084/m9.figshare.5901640. Relevant data are also available from the authors upon request.

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

## Acknowledgements

This work was supported by the Deutsche Forschungsgemeinschaft (DFG), Grant No. BR 5221/1-1. Numerical calculations were performed in part with resources supported by the Zentrum für Datenverarbeitung of the University of Tübingen.

## Author contributions

D.B. initiated the idea and L.F. made the calculations and numerical simulations. Both authors contributed to the interpretation of data and the writing of the manuscript.

## Additional information

**Competing interests:** The authors declare no competing interests.

