## [Peer Review File · Nature Communications]

Reviewers' comments:

Reviewer #1 (Remarks to the Author):

The request for review was received at 28 March 2017

The report is sent to Editor on 1 March 2017

Global remarks:

A) The introduction part (p.2-3) is rather technical, formal and does not present well enough the physics of the system

B)Sec. System-Results:

it is not clearly explained what is the system which

we want to detect and what is the sensor;

it would be natural to study a qubit connected

to a kicked top but

such cases had been already studied

(e.g. J.W.Lee et al Phys Rev A 72, 012310 (2005)

with a qubit connected to a chaotic system

and Refs. therein)

C) From the fidelity behavior in quantum systems

(see e.g. P.Jacquod et al Phys Rev E 64, 055203(R) (2001),

G.Benenti and G.Casati Phys Rev E 65, 066205 (2002))

it is known that a small coupling

(or small perturbation strength for fidelity decay)

behaves like a usual golden mean transition rate;

so small perturbations will not produce something remarkable for sensor detection; in contrast for strong perturbations one can find the Lyapunov regime typical for quantum chaos but such strong perturbations will modify a system we want to measure/detect.

Since the paper does not present clearly the links between qubit (we want to detect) and quantum chaotic system (sensor) it is not clear if a strong coupling regime would allow to measure a system without strongly perturbing it.

D)The presented results (e.g. Fig.1) correspond to unrealistically strong momentum j values which are not suitable for realistic experiments with molecules having j about 20.

Technical notes:

page2: POVM is not defined

Due to above reasons I do not consider this paper to be suitable for publication in Nature Communications.

Reviewer #2 (Remarks to the Author):

In this paper, the authors propose and test an idea that aims at enhancing precision in the problem of quantum parameter estimation. Their idea is to use quantum chaos—the hypersensitivity of certain quantum systems to perturbations to their Hamiltonians—to achieve this task. In principle, this seems to be a reasonable approach: The system one chooses should be sensitive to the parameter one is interested to measure, and the dynamics should be deterministic. Both premises are found in the Kicked Top system, which the authors have chosen as their system of study. In this sense, the hypersensitivity to small perturbations in the system Hamiltonian can be thought of as a feature when one is interested in amplifying the effect of a small change in the parameter one measures in practice.

Traditionally, as the authors correctly point out in their manuscript, enhancements in metrology are achieved by using highly entangled states as the main resource in most protocols. This path is certainly a complicated one since creating and maintaining entanglement in practical applications is no easy task. The chaos-based approach, however, does not rely on such entangled states. In fact, using simple-to-prepare spin coherent states seems to suffice to already see effects that beat the standard quantum limit (SQL), according to the results presented in the manuscript.

Specifically, the authors propose using the kicked top system (realizable in ensembles of cold atoms, for example) to do magnetometry. Their objective is to determine the parameter α (proportional to the magnitude of an external magnetic field) by making the system evolve via the standard Floquet dynamics of the Kicked Top. They look at the Cramer-Rao bound and thus they use the Fisher information as the figure of merit to assess the performance of their method. All this is done as numerical experiments. The authors show several plots in which it is clear that the Fisher information of the measured parameter α strongly depends on whether the system is chaotic or not and describe the the areas of phase space in which one would like to be in order to achieve high sensitivity. Moreover, the authors study the kicked top metrology protocol under a decoherence model and show that it still outperforms a non-chaotic top under the same model.

Without decoherence, the results show that for shorter times, the chaotic dynamics is an advantage. However, for longer times, the scaling is similar to that of a simple top and there's no advantage of having a chaotic system. For a model with decoherence, the results are similar, however, the advantage seems to be smaller.

In principle, I consider this to be an interesting idea and a plausible one in practical cases. However, there are a few concerns that must be addressed before I make my final recommendation.

Questions and comments:

1. A major concern that I have related to using dynamical systems to extract information about a system in practice is the question of robustness to experimental errors. The authors have done a deep study in the case of decoherence, but it would be interesting to know what happens if the parameters of the nonlinear term in the Hamiltonian are slightly perturbed or if the measurement basis is slightly rotated. Would the enhancement in measuring the parameter α still be present?

My guess is that the same chaotic behavior, which produces the enhancement in the measurement in the first place, will be very difficult to control if, say, k is not exactly known, and the enhancements will be greatly decreased.

2. In the last paragraph of section II (just before the discussion section), the authors discuss for the first time the measurements they envision making to achieve their chaos-enhanced parameter estimation scheme. Because of the importance of this issue, it seems that the discussion should be done in more depth. Can the authors comment more on this? And also, on how the measurement discussed relates to Fig. 5 and current metrology schemes. The authors conclude that their scheme is superior to current magnetometers, but I failed to see a proper comparison.

3. In this context, the purpose of Fig. 5 is not entirely clear to me. Can the authors clarify this? Also, I_{Fisher} should be defined before the figure appears or at least commented. The definition appears at the end of the paper in the methods section, but I find this to be suboptimal. Also, why is the plot 5b seemingly showing no-damping dynamics?

4. When damping is included in the study, it is not clear to me how the chaos dynamics compares to the SQL. Such discussion was very prominent in the damping-free study. Can the authors comment on this and include a discussion about it in the main text?

5. Eqs. 8 and 9: The Floquet map is computed for a period τ . Is $t = n\tau$? where n is an integer number of applications of U ? Please clarify. The notion of time should be discussed more prominently after these equations.

6. Is there any reason the authors, when studying the problem of the damped top, only present results for $j = 40$ in Fig. 4? What's the limitation for going higher, as they did in the non damped case or in later plots?

7. Most of the time, the Floquet map is written as $U_{\alpha}(k)$, however on page 5, just above eq. (8) it is written as $U(\alpha,k)$. Please make the notation consistent.

8. This could be trivial, but there is something odd with the notation specially in eqs. (8) and (9). In the left-hand side, we have an α dependence while in the right-hand side there is none. Can the authors make this more explicit? Maybe saying the equations are valid for all α would be enough.

9. The gain G uses the same symbol as the Hermitian generator defined before eq. (3). Please use different symbols for different quantities.

10. In Fig. 1, there is the quantity $R_z(\alpha)$ that seems to be the definition of the rotation around z by an angle α , but this has to be stated clearly.

11. Just before Fig. 2, the Gain is defined in terms of a mysterious quantity H , which should be I , the Fisher information. Please fix this.

12. Just before Fig. 2, the parameter t is set to 2^{15} . What does that mean? 2^{15} kicks in units of unit τ ? Please clarify.

13. In eq. (11), please define J_+ , J_-

14. Define U in eq. (12) if different from the Floquet operator. If not, use $U_{\alpha}(k)$.

15. In Fig. 5a, H should be defined if different from I .

Reviewer #3 (Remarks to the Author):

This is an interesting piece of work. The basic premise follows from the idea that the same initial wave function, propagated by evolution operators governed by two slightly different Hamiltonians,

can have an evolution which is very sensitive to the differing values of parameters in the Hamiltonians, if the evolution of the Hamiltonians is expected to be chaotic in the classical limit. The way that the authors describe this is through the expectation value with regard to the initial state of the "echo-operator", but which is actually equivalent to the overlap in the final wave functions propagated by the two slightly different evolutions. This in turn can be related to the quantum fisher information, generally accepted as a good measure of "goodness" of a sensing protocol.

Although the investigation has a fairly concrete motivation, it is nonetheless more a somewhat abstract investigation into the general idea of how such sensitivity could be useful for sensing devices in general, using the top, kicked top, dissipative top, dissipative kicked top as a general example of how this might work in principle, without assuming too many specifics (the comment is made, e.g., that "one needs also to specify the actual measurement of the probe") (whether J_x , J_y , J_z , or some combination thereof, in the case of the kicked tops considered here). Something which also reinforces this emphasis in this paper is also the very large values of t and j considered (corresponding to hundreds of thousands of kicks, and thousands of angular momentum states) -- while interestingly illustrative of trends, these would certainly be a challenge in an atomic physics experiment, for example! On the other hand, the authors do remark (in the discussion) that significant enhancement of the QFI is observed also for small values of j , and reference is made to the collective pseudo spin of a system of N atoms couple to a common cavity mode in appendix B.

I don't necessarily object to a comparatively abstract investigation into the idea of exploiting this sensitivity, and am indeed inclined to view this work sympathetically, however i do think the authors should be a bit more up front and clear that this is actually what they're setting out to do, rather than making a reasonably specific proposal for a particular class of experiments. I have one specific question which I'd like the authors to consider -- namely, is it necessary for the system to be quantum chaotic, or is it sufficient to consider a system, the classical limit of which has a hyperbolic fixed point (such an "inverted harmonic oscillator" type Hamiltonian $H = p^2 - \alpha x^2$), which also has a positive Lyapunov exponent?

There are also a number of minor comments, which I think should be addressed

Minor notes:

1. "Unitary" is not a noun (should be "a slightly perturbed unitary operator", for example)
2. I'm a bit puzzled by the notation used to describe the kicked top -- in the main text KT is written with a calligraphic K and an italic T, as a subscript in mathematical formulae it is a standard roman "KT". This is a bit inconsistent (and the use of two different fonts in one abbreviation is a bit bizarre DT and DKT also have this in the paper)

3. The pseudo in "(pseudo-)angular" isn't explained (I presume it's supposed to convey that it doesn't literally have to be angular momentum, so long as the algebraic properties basically correspond, but this should be clarified)
4. The time ordering operator is undefined when it is first introduced (just above Eq. (4)), and sometimes has a hat on it, and sometimes doesn't.
5. Below Eq (9) there's a statement describing N^2 type scaling. This is a bit cryptic -- presumably N is supposed to be some measure of the size of the system, to be conveyed by j , in this case.

However, I find the general research to be interesting, creative, and reasonably well explained. Assuming all of the above comments are satisfactorily addressed, I would think publication would be quite acceptable.

Reviewer #4 (Remarks to the Author):

In this work, the authors propose chaotic systems for quantum-enhanced metrology. They study the kicked-top, a canonical model in the study of chaos, and conclude that it can help improve the sensitivity of existing magnetic field sensors. They also study a dissipative version. While the thought is an interesting one, there are several gaps in the arguments that render this paper unpublishable in Nature Communications. I enlist them below

`\begin{enumerate}`

`\item` The authors suggest that their method surpasses the SQL without using quantum entanglement. It is known that entanglement is necessary to beat the SQL for a unitary parameter, and indeed only 2-body entanglement is sufficient (Baumgratz, T. & Datta, A. Quantum Enhanced Estimation of a Multidimensional Field. Phys. Rev. Lett. 116, 30801 (2016).). The hamiltonian that authors use in Eq. 5 evidently produces entanglement. So, why the claim "Many ways of breaking the SQL without using entanglement have been explored (see [18] for a recent review). Here we investigate yet another possibility ..."? Indeed, the authors should clarify how much entanglement their scheme generates.

`\item` To attain the quantum limits, the authors propose a $\$J_x\$$ (or $\$J_y\$$) measurement, and show how it outperforms the benchmark for some values of k in Fig. 5. For most values of k , these measurements are poorer than the benchmark scheme. So, the authors have shown that most of

the time, their measurements don't work. Unless there is a POVM that beats the benchmark all (or most of) the time, this proposal is useless.

\item The authors never discuss how their scheme is better than a corresponding classical scheme. A classical chaotic system is also exponentially sensitive to initial conditions (and maybe parameters). So, its unclear to me what is quantum about this whole scheme?

\item Speaking of which, why is it that all classical sensors are not based on chaotic systems if they are so sensitive? If they are not, what makes the authors think that they will be any use in the quantum case ?

\item Since chaotic systems are exponentially sensitivity, why don't the sensors the authors propose offer exponential precision?

\end{enumerate}

In conclusion, this paper should be rejected.

Answer to the questions and remarks of the Reviewers

Reviewer #1

Global remarks:

A) The introduction part (p.2-3) is rather technical, formal and does not present well enough the physics of the system

We rewrote and restructured the introduction, which now contains no more formulas at all. We also describe in words the considered system already in the introduction now, and make clear in particular that a spin is used as sensor and the magnetic field to be measured is classical.

B)Sec. System-Results:

it is not clearly explained what is the system which we want to detect and what is the sensor; it would be natural to study a qubit connected to a kicked top but such cases had been already studied (e.g. J.W.Lee et al Phys Rev A 72, 012310 (2005) with a qubit connected to a chaotic system and Refs. therein)

Our work is not about detecting a system, but measuring a classical magnetic field. Extremely small magnetic fields are not necessarily quantum in nature, but arise in many fields such as geophysics, medical diagnostic, material testing and many more. The origin of the field is irrelevant for our concerns as long as the field can be considered a given classical parameter, which is a standard assumption in magnetometry.

The paper of J.W. Lee et al. studies decoherence of a single qubit coupled to a chaotic, quasi-classical system, with the idea of obtaining a detector of the state of the qubit. It has nothing to do with magnetometry or the increase of precision of the measurement of a classical parameter by using quantum chaos.

In the new version we made very clear already in the introduction that we are interested in measuring a *classical* magnetic field.

C) From the fidelity behavior in quantum systems (see e.g. P.Jacquod et al Phys Rev E 64, 055203(R) (2001), G.Benenti and G.Casati Phys Rev E 65, 066205 (2002)) it is known that a small coupling (or small perturbation strength for fidelity decay) behaves like a usual golden mean transition rate; so small perturbations will not produce something remarkable for sensor detection; in contrast for strong perturbations one can find the Lyapunov regime typical for quantum chaos but such strong perturbations will modify a system we want to measure/detect. Since the paper does not present clearly the links between qubit (we want to detect) and quantum chaotic system (sensor) it is not clear if a strong coupling regime would allow to measure a system without strongly perturbing it.

This remark is based once more on the erroneous assumption that we want to measure/detect a quantum system, such as a single spin. As explained above (and made clearer in the new version), our work is about measuring a classical magnetic field, where by definition no back action from the sensor is taken into account.

The second misunderstanding is that the coupling strength relevant for the fidelity decay is given by the coupling of the detector to the magnetic field. The “perturbation” that is really relevant for the quantum Fisher information is the difference in the Hamiltonians with two infinitesimally different values of the magnetic field. Hence, in the language of the Jacquod et al. paper, one is always in the perturbative regime, which in fact leads to a Gaussian decay of fidelity with time, and in the limit of vanishing perturbation to a power law decay of fidelity, alias increase of QFI, see eqs. (10,11) in the new version.

More generally, while it is true that the rapid decay of fidelity in a quantum chaotic system upon change of a parameter presents a more complex (but well-known) picture than what we cannot fully review in the present article (we cite the review by Gorin et al. instead), it is only the initial idea and motivation for studying quantum-chaotic sensors. Our results clearly demonstrate that the method works, so there is no point in arguing about the motivation. The claim of the Reviewer “so small perturbations will not produce something remarkable for sensor detection;” contradicts the massive amount of results presented (assuming that he/she means with “something remarkable” an increased sensitivity).

To give the interested reader direct access to the original papers studying the fidelity decay (rather than through the review article which we cited already in the 1st version), we have added the two references cited by the Reviewer.

D) The presented results (e.g. Fig.1) correspond to unrealistically strong momentum j values which are not suitable for realistic experiments with molecules having j about 20.

Realistic spin sizes depend on the studied system. Besides the spin $j=2$ studied already in the 1st version, we now investigate also a rather small atomic spin $j=3$ in the context of the SERF magnetometer and still find a substantial enhancement. But as we describe in the new version, much larger spins are available, for instance, by exploiting the collective spin of an ensemble of atoms ($j \gg 200$). This needs a coupling between the spins for realizing the non-linear term, but how to introduce such interactions is known from spin-squeezing physics. Therefore, the scaling of measurement precision with increasing j is of interest and may motivate further experimental work targeted at applying spin-squeezing techniques to kicking.

Technical notes:

page2: POVM is not defined

We now spell out the abbreviation: positive-operator valued measure (POVM) and define a POVM by positivity and completeness of the POVM elements.

Due to above reasons I do not consider this paper to be suitable for publication in Nature Communications.

We hope that our replies and the improvements in the new version convince Reviewer 1 to change his/her mind.

Reviewer #2:

In this paper, the authors propose and test an idea that aims at enhancing precision in the problem of quantum parameter estimation. Their idea is to use quantum chaos—the hypersensitivity of certain quantum systems to perturbations to their Hamiltonians—to

achieve this task. In principle, this seems to be a reasonable approach: The system one chooses should be sensitive to the parameter one is interested to measure, and the dynamics should be deterministic. Both premises are found in the Kicked Top system, which the authors have chosen as their system of study. In this sense, the hypersensitivity to small perturbations in the system Hamiltonian can be thought of as a feature when one is interested in amplifying the effect of a small change in the parameter one measures in practice.

Traditionally, as the authors correctly point out in their manuscript, enhancements in metrology are achieved by using highly entangled states as the main resource in most protocols. This path is certainly a complicated one since creating and maintaining entanglement in practical applications is no easy task. The chaos-based approach, however, does not rely on such entangled states. In fact, using simple-to-prepare spin coherent states seems to suffice to already see effects that beat the standard quantum limit (SQL), according to the results presented in the manuscript.

Specifically, the authors propose using the kicked top system (realizable in ensembles of cold atoms, for example) to do magnetometry. Their objective is to determine the parameter α (proportional to the magnitude of an external magnetic field) by making the system evolve via the standard Floquet dynamics of the Kicked Top. They look at the Cramer-Rao bound and thus they use the Fisher information as the figure of merit to assess the performance of their method. All this is done as numerical experiments. The authors show several plots in which it is clear that the Fisher information of the measured parameter α strongly depends on whether the system is chaotic or not and describe the the areas of phase space in which one would like to be in order to achieve high sensitivity. Moreover, the authors study the kicked top metrology protocol under a decoherence model and show that it still outperforms a non-chaotic top under the same model.

Without decoherence, the results show that for shorter times, the chaotic dynamics is an advantage. However, for longer times, the scaling is similar to that of a simple top and there's no advantage of having a chaotic system. For a model with decoherence, the results are similar, however, the advantage seems to be smaller.

In principle, I consider this to be an interesting idea and a plausible one in practical cases. However, there are a few concerns that must be addressed before I make my final recommendation.

Questions and comments:

1. A major concern that I have related to using dynamical systems to extract information about a system in practice is the question of robustness to experimental errors. The authors have done a deep study in the case of decoherence, but it would be interesting to know what happens if the parameters of the nonlinear term in the Hamiltonian are slightly perturbed or if the measurement basis is slightly rotated. Would the enhancement in measuring the parameter α still be present?

My guess is that the same chaotic behavior, which produces the enhancement in the measurement in the first place, will be very difficult to control if, say, k is not exactly known, and the enhancements will be greatly decreased.

We thank the Reviewer for his/her insightful and highly relevant questions and comments, and are happy that he/she thinks that our scheme is an “interesting idea and a plausible one in practical cases”.

We addressed question 1 in the new version through numerical experiment: In a concrete example with a specified measurement we investigate a slight perturbation of the kicking strength by calculating a Fisher information which takes into account a 5% variance in Gaussian distributed k values and find that Fisher information reduces only marginally. This is a typical error estimate for the kicking strength given in the paper by Chaudhury et al. on the experimental realization of the kicked top. While as the Reviewer expected the sensitivity is reduced, the advantage over the reference persists (see Fig. 6 in the new version).

2. In the last paragraph of section II (just before the discussion section), the authors discuss for the first time the measurements they envision making to achieve their chaos-enhanced parameter estimation scheme. Because of the importance of this issue, it seems that the discussion should be done in more depth. Can the authors comment more on this? And also, on how the measurement discussed relates to Fig. 5 and current metrology schemes. The authors conclude that their scheme is superior to current magnetometers, but I failed to see a proper comparison.

We expanded the part based on actual measurements (rather than QFI) by providing now two examples for the kicked top (see Fig. 6), one with and one without decoherence. Furthermore, we present in the new version an in-depth analysis of a state-of-the-art atomic-vapor magnetometer in the SERF regime. With this we are able to not only to show unambiguously the superiority of our scheme with respect to the standard SERF magnetometers but to also provide another example of a measurement, namely the measurement of an electron-spin component. This is very close to readout schemes that use the Faraday rotation of the polarization of a probe beam, proportional to the expectation value of a component of the atomic spin.

3. In this context, the purpose of Fig. 5 is not entirely clear to me. Can the authors clarify this? Also, I_{Fisher} should be defined before the figure appears or at least commented. The definition appears at the end of the paper in the methods section, but I find this to be suboptimal. Also, why is the plot 5b seemingly showing no-damping dynamics?

We now define Fisher information in the beginning of the Results section. We replaced plot 5b (which indeed was not very clear but actually showed dynamics with damping) by Fig. 6 which presents another measurement with and without damping.

4. When damping is included in the study, it is not clear to me how the chaos dynamics compares to the SQL. Such discussion was very prominent in the damping-free study. Can the authors comment on this and include a discussion about it in the main text?

In the presence of dissipation the SQL is not an appropriate benchmark anymore, as even without kicking it can typically not be reached anymore. In addition, numerical constraints limit j in this case to values up to 200, i.e. a scaling behaviour as function of j will not be very useful over such a limited range. Instead, we use the non-kicked dissipative top as a benchmark, see the discussion on the bottom of p. 11 in the new version, which we believe presents the most direct assessment of the usefulness of our method.

5. Eqs. 8 and 9: The Floquet map is computed for a period τ . Is $t = n\tau$? where n is an

integer number of applications of U ? Please clarify. The notion of time should be discussed more prominently after these equations.

This is correct, $t=n*\tau$. We clarified this in the paper and use t and τ consistently now.

6. Is there any reason the authors, when studying the problem of the damped top, only present results for $j = 40$ in Fig. 4? What's the limitation for going higher, as they did in the non damped case or in later plots?

We chose $j=40$ for presenting the typical behaviour of QFI in the presence of damping. There is no particular reason we chose $j=40$. We now write "...To illustrate the typical behaviour of QFI we exemplarily choose certain spin sizes j and damping constants κ here and in the following..."

The limitation to go much higher than $j=200$ comes from the fact that we calculate the dissipative propagator from an analytical solution that, evaluated with the necessary numerical precision, takes more and more time with increasing j .

7. Most of the time, the Floquet map is written as $U_{\alpha}(k)$, however on page 5, just above eq. (8) it is written as $U(\alpha,k)$. Please make the notation consistent.

We corrected the notation which is now consistent.

8. This could be trivial, but there is something odd with the notation specially in eqs. (8) and (9). In the left-hand side, we have an α dependence while in the right-hand side there is none. Can the authors make this more explicit? Maybe saying the equations are valid for all α would be enough.

We now say that it holds for all α .

9. The gain G uses the same symbol as the Hermitian generator defined before eq. (3). Please use different symbols for different quantities.

We now use the symbol Γ for the gain.

10. In Fig. 1, there is the quantity $R_z(\alpha)$ that seems to be the definition of the rotation around z by an angle α , but this has to be stated clearly.

We state it clearly now in the caption of Fig. 1.

11. Just before Fig. 2, the Gain is defined in terms of a mysterious quantity H , which should be I , the Fisher information. Please fix this.

We fixed this and now use the symbol I for the QFI and I_{Fisher} for the (classical) Fisher information.

12. Just before Fig. 2, the parameter t is set to 2^{15} . What does that mean? 2^{15} kicks in units of unit τ ? Please clarify.

Yes, $t=2^{15}$ means 2^{15} time steps each of length τ and each involving an instantaneous kick at the end.

This is just an example where we chose some large value for t which must be set to a value to plot the data in the way we did. We clarify the notion of time in the sentence with eq. 2.

13. In eq. (11), please define J_{+} , J_{-}

We now give a definition J_{+} and J_{-} .

14. Define U in eq. (12) if different from the Floquet operator. If not, use $U_{\alpha}(k)$.

We now use $U_{\alpha}(k)$.

15. In Fig. 5a, H should be defined if different from I .

We replaced H with I .

We thank Reviewer 2 for his/her constructive criticism, which helped us improve the paper considerably. We hope that he/she is convinced now that the paper is ready for publication in Nature Communications.

Reviewer #3:

This is an interesting piece of work. The basic premise follows from the idea that the same initial wave function, propagated by evolution operators governed by two slightly different Hamiltonians, can have an evolution which is very sensitive to the differing values of parameters in the Hamiltonians, if the evolution of the Hamiltonians is expected to be chaotic in the classical limit. The way that the authors describe this is through the expectation value with regard to the initial state of the "echo-operator", but which is actually equivalent to the overlap in the final wave functions propagated by the two slightly different evolutions. This in turn can be related to the quantum fisher information, generally accepted as a good measure of "goodness" of a sensing protocol.

Although the investigation has a fairly concrete motivation, it is nonetheless more a somewhat abstract investigation into the general idea of how such sensitivity could be useful for sensing devices in general, using the top, kicked top, dissipative top, dissipative kicked top as a general example of how this might work in principle, without assuming too many specifics (the comment is made, e.g., that "one needs also to specify the actual measurement of the probe") (whether J_x , J_y , J_z , or some combination thereof, in the case of the kicked tops considered here). Something which also reinforces this emphasis in this paper is also the very large values of t and j considered (corresponding to hundreds of thousands of kicks, and thousands of angular momentum states) -- while interestingly illustrative of trends, these would certainly be a challenge in an atomic physics experiment, for example! On the other hand, the authors do remark (in the discussion) that significant enhancement of the QFI is observed also for small values of j , and reference is made to the collective pseudo spin of a system of N atoms couple to a common cavity mode in appendix B.

I don't necessarily object to a comparatively abstract investigation into the idea of exploiting this sensitivity, and am indeed inclined to view this work sympathetically, however i do think the authors should be a bit more up front and clear that this is actually what they're setting out to do, rather than making a reasonably specific proposal for a particular class of experiments.

We thank the Reviewer for his interest and his sympathetic view of our paper.

While we were ready to be more up front that so far we had presented rather a proof of principle than a specific proposal for a particular class of experiments, the editor's advice was to do the latter. So we now present in addition to the results from the kicked top those from a detailed and established model of a SERF atomic-vapor magnetometer including all relevant decoherence processes. These investigations can serve as a blue print for an actual atomic physics experiment, including values of the experimental parameters such as laser intensities, detunings etc., summarized in Supplementary Note 2.

We also give a perspective how to increase the effective spin size in this concrete example.

I have one specific question which I'd like the authors to consider -- namely, is it necessary for the system to be quantum chaotic, or is it sufficient to consider a system, the classical limit of which has a hyperbolic fixed point (such an "inverted harmonic oscillator" type Hamiltonian $H = p^2 - \alpha x^2$), which also has a positive Lyapunov exponent?

We thank the Reviewer for this interesting idea. Indeed one might expect that an inverted harmonic oscillator presents the advantage of amplification compared to a regular one. It is not clear, however, how much of this advantage would persist in the presence of decoherence. The kicking approach in our paper has the advantage that coherences are being re-created over and over again, and in the limit of large times prevent the system from relaxing to a useless thermal state. The latter property would be absent in an autonomous system. Altogether, investigating the system proposed by the Reviewer is beyond the scope of the present work, which is already at the limit of what can be reasonably explained within the length limitations set by the journal. We preferred to use the remaining space and resources to investigate the realistic model of a SERF magnetometer, as suggested by the editor.

There are also a number of minor comments, which I think should be addressed

Minor notes:

1. "Unitary" is not a noun (should be "a slightly perturbed unitary operator", for example)

We now use "unitary" as an adjective.

2. I'm a bit puzzled by the notation used to describe the kicked top -- in the main text KT is written with a calligraphic K and an italic T , as a subscript in mathematical formulae it is a standard roman "KT". This is a bit inconsistent (and the use of two different fonts in one abbreviation is a bit bizarre DT and DKT also have this in the paper)

We now use calligraphic D, K, T consistently. We keep a different font, however, for labelling the state considered (e.g. CS for coherent state), which we think reasonable for distinguishing between system and state.

3. The pseudo in "(pseudo-)angular" isn't explained (I presume it's supposed to convey that it doesn't literally have to be angular momentum, so long as the algebraic properties basically correspond, but this should be clarified)

Yes. We added a sentence after the introduction of the kicked top in order to clarify this.

4. The time ordering operator is undefined when it is first introduced (just above Eq. (4)), and sometimes has a hat on it, and sometimes doesn't.

We define the time ordering in the new version and omit the hat on it consistently.

5. Below Eq (9) there's a statement describing N^2 type scaling. This is a bit cryptic -- presumably N is supposed to be some measure of the size of the system, to be conveyed by j , in this case.

We now write "SQL-type scaling with $N=2j$, when the spin- j is composed of N spin- $\frac{1}{2}$ particles in a state invariant under permutations of particles. "

However, I find the general research to be interesting, creative, and reasonably well explained. Assuming all of the above comments are satisfactorily addressed, I would think publication would be quite acceptable.

We are grateful for this judgement and the constructive criticism of Reviewer 3. We hope that we have addressed all comments satisfactorily.

Reviewer #4:

In this work, the authors propose chaotic systems for quantum-enhanced metrology. They study the kicked-top, a canonical model in the study of chaos, and conclude that it can help improve the sensitivity of existing magnetic field sensors. They also study a dissipative version. While the thought is an interesting one, there are several gaps in the arguments that render this paper unpublishable in Nature Communications. I enlist them below

`\begin{enumerate}`

`\item` The authors suggest that their method surpasses the SQL without using quantum entanglement. It is known that entanglement is necessary to beat the SQL for a unitary parameter, and indeed only 2-body entanglement is sufficient (Baumgratz, T. & Datta, A. Quantum Enhanced Estimation of a Multidimensional Field. Phys. Rev. Lett. 116, 30801 (2016).). The hamiltonian that authors use in Eq. 5 evidently produces entanglement. So, why the claim "Many ways of breaking the SQL without using entanglement have been explored (see [18] for a recent review). Here we investigate yet another possibility ..."? Indeed, the authors should clarify how much entanglement their scheme generates.

The statement in the cited PRL that entanglement is necessary for breaking the SQL is based on Hamiltonian (3) in that paper, a sum of phase shift Hamiltonians. Our Hamiltonian is clearly not of that form, due to the additional non-linear, parameter-independent term. In addition it is time-dependent, and our scheme hence breaks with the traditional paradigm of initial state preparation, imprinting the parameter through a parameter-dependent unitary evolution driven by a time-independent Hamiltonian, and read-out. Hence, there is no reason that the results of the cited PRL should apply.

In addition, statements about the importance of entanglement in quantum-enhanced measurements are about the entanglement present at the moment of encoding the parameter, created in the step of initial state preparation. This is also the understanding of the necessary correlations in the cited PRL. The coherent states that we use as initial state are clearly not

entangled, and whether entanglement is created through the parameter coding evolution itself is irrelevant in the framework of the traditional paradigm.

The Reviewer is right that the Hamiltonian of the kicked top will create entanglement – if decoherence is not too strong (note that decoherence was not considered in the cited proof). But it cannot explain the enhancement of sensitivity, as we are outside the mentioned paradigm, and hence it does not make sense for us to quantify the entanglement during the time evolution.

We realize, however, that claiming that the scheme does not rely on entanglement creates unnecessary controversy, and hence removed the corresponding sentences. The only claim made now is that the initial state is free from entanglement, which is definitely true. And what really counts is that the method works.

\item To attain the quantum limits, the authors propose a J_x (or J_y) measurement, and show how it outperforms the benchmark for some values of k in Fig. 5. For most values of k , these measurements are poorer than the benchmark scheme. So, the authors have shown that most of the time, their measurements don't work. Unless there is a POVM that beats the benchmark all (or most of) the time, this proposal is useless.

We strongly disagree that the proposal is useless because there are parameter regimes where the benchmark without kicking works better. The kicking strength k is a new degree of freedom that we can freely adjust to give an improved sensitivity. We have demonstrated that there are broad intervals of kicking strength where an improved sensitivity is found, so there is no issue with fine tuning, and in the new version we even demonstrate that randomly varying kicking strength with a reasonable (and experimentally realized) spread of 5% still allows one to outperform the benchmark.

\item The authors never discuss how their scheme is better than a corresponding classical scheme. A classical chaotic system is also exponentially sensitive to initial conditions (and maybe parameters). So, it's unclear to me what is quantum about this whole scheme?

What is quantum about the system is the fact that we have a quantum angular momentum with quantum dynamics described by Schrödinger's equation, leading to the propagation of a state vector in Hilbert space, or more generally a density matrix. The spin shows quantum noise and is capable of quantum interference, things all absent for a classical kicked top. Judging from the scaling of the QFI with j and the fact that the classical limit corresponds formally to $j \rightarrow \infty$ (even though this is only one aspect of the classical limit; one also has to deal with the coherences that vanish in the classical case of propagation in phase space), one might indeed speculate that also a classical system might allow enhanced precision when rendering it chaotic. But we don't make any statement about it, as this is beyond the scope of the present work, and such a possibility does not invalidate our work.

\item Speaking of which, why is it that all classical sensors are not based on chaotic systems if they are so sensitive? If they are not, what makes the authors think that they will be any use in the quantum case ?

There are in fact classical sensors based on unstable systems in a broader sense, such as photon-detectors based on a super-conducting to normal phase transition, or systems tuned to a phase transition more generally. Also, each amplifier is in a sense an unstable system, amplifying a small input to a large output. But what makes us think that chaotic systems are

of use for metrology in the quantum case is, of course, the wealth of hard evidence based on numerical and analytical results that we present.

\item Since chaotic systems are exponentially sensitivity, why don't the sensors the authors propose offer exponential precision?

It is difficult to answer this question without knowing what Reviewer 3 means with “exponential precision”: Exponential in what?

Our results show a power law increase of the QFI with time and system size which is in sync with the known results about the Loschmidt echo in the limit of infinitesimally small change in the parameter. Due to the infinitesimally small perturbation relevant for the QFI, one is always in the perturbative regime, in which finite small perturbation lead to a Gaussian decay of fidelity with time that goes over into the mentioned power laws for the infinitesimally small changes. We added two sentences in the new version after eq. 11 that explain this. Nevertheless, we demonstrate that the QFI correlates very well with the Lyapunov exponent of classical trajectories starting at the location of the chosen initial coherent states. Hence, a signature of the corresponding classical trajectories diverging exponentially in time is still present.

\end{enumerate}

In conclusion, this paper should be rejected.

We disagree, of course. Besides proposing a new idea that opens up a novel route out of the current paradigm in quantum metrology and which now can be exploited for many other systems, we also show in the new version by rather detailed and realistic simulations of a SERF magnetometer that the precision of these devices that already now count amongst the most sensitive existing, can be further improved by non-linear kicks. Given the importance of high-precision magnetometry for many fields in science, we believe that this is reason enough to attract the attention of the broad readership of Nature Communications.

Reviewers' comments:

Reviewer #1 (Remarks to the Author):

In my opinion the chaotic sensor is interesting to measure qubit, but now the authors state that this is not their aim.

In this view the work is not of high level interest of NatCom.

I propose that the authors address it to other journal.

The results of Lee et al. Phys. Rev. A v.72, p.012310 (2005)

clearly show that only strong coupling of a qubit to a quantum chaotic sensor (detector)

can significantly modify a detector state in such a way that

a clear detector signal is well visible differently

for two states of qubit (up/down).

At weak coupling quantum chaos has a modification of detector state but since it is chaotic there is no clear difference for qubit states being up or down.

This feature is missed by the authors.

If we will measure another object (e.g. a large moment instead of qubit) a similar situation will appear as for a qubit.

Due to these reasons I think that the paper does not correspond to high standards of NatCom.

Reviewer #2 (Remarks to the Author):

The authors have reviewed and addressed all of my previous concerns. Specifically, they now explain that their method is robust to realistic variations of the system parameters while still retaining the advantages for metrology of magnetic fields. The idea of using a dynamical system which is not integrable as a sensor is very interesting and I believe the authors' analysis show in principle that it could eventually be applied in reality. With all the changes and the new plots, I think this paper should be accepted for publication.

Reviewer #3 (Remarks to the Author):

In summary, I am basically satisfied with the authors' response to my report. They note that they have chosen to present a fairly specific model as an example of their general point, and I am happy with this as a response, even though it is a bit different to what I had written (in fact I think it is better).

As such I am happy to say that I think the manuscript in its present form is publishable in Nature Communications.

Just a very slight note --- in the abstract it should be "kicks render" rather than "kicks renders."

Answer to the questions and remarks of the Reviewers

Reviewer #1

In my opinion the chaotic sensor is interesting to measure qubit, but now the authors state that this is not their aim.

We never stated that we intend to measure a qubit. We want to measure a classical magnetic field which represents the typical setup in magnetometry. This was clear already in the first version. The paper did not even contain the word qubit, and a single look at the Hamiltonian reveals that the magnetic field is a classical parameter in the model.

In this view the work is not of high level interest of NatCom.

We do not agree. We consider magnetometry a topic of high level interest which is reflected by the wealth of papers published on this topic: New world records or techniques for measuring small magnetic fields regularly make their way to Nature journals, see e.g. ^{1,2,3,4,5} to Physical Review Letters ^{6,7,8,9} or Physical Review X ¹⁰ to name a few. Furthermore, the field of high-precision measurements of magnetic fields is a billion dollar industry (see e.g. http://www.strategyr.com/MarketResearch/Magnetic_Sensors_Market_Trends.asp) whose products have huge impact in many other fields of science and medicine.

I propose that the authors address it to other journal.

As shown above, the paper will be in good company there, and we therefore consider Nature Communications a proper journal to publish our work.

The results of Lee et al. Phys. Rev. A v.72, p.012310 (2005) clearly show that only strong coupling of a qubit to a quantum chaotic sensor (detector) can significantly modify a detector state in such a way that a clear detector signal is well visible differently for two states of qubit (up/down). At weak coupling quantum chaos has a modification of detector state but since it is chaotic there is no clear difference for qubit states being up or down. This feature is missed by the authors.

We are perplex that the Reviewer still tries to squeeze the paper into the context of detecting the state of a qubit after having realized that the paper is *not* about detecting the state of a qubit. He /she insists that a strong coupling would be needed to distinguish the two states, a point that according to him/her we have overlooked ("This feature is missed by the authors."), while a few lines before he/she realized that we don't even have a qubit to couple to! This does not make any sense.

If we will measure another object (e.g. a large moment instead of qubit) a similar situation will appear as for a qubit.

This is too generic a statement to be answered precisely. Presumably the Reviewer tries to imply with this that we could not measure a classical magnetic field more precisely either. But this is totally ignoring all the evidence that we present. A single look at one of the figures, e.g. Fig. 1d, shows that a large enhancement of sensitivity can result from the non-linear dynamics. Secondly, even if the Reviewer's generic single-line statement "a similar situation will appear as for a qubit" was correct, the analogy would be about measuring the state of that larger spin, not measuring its

magnetic field. These are very different tasks as is already obvious from the fact that even a large spin has only a finite number of possible eigenstates of J_z , whereas a classical magnetic field varies continuously.

Due to these reasons I think that the paper does not correspond to high standards of NatCom.

We disagree and believe there is no point in arguing scientifically any further against the objections of Reviewer 1 as he/she still seems not to understand what the paper is about, simply ignores presented data, and has strangely biased ideas about what is of “high level interest” for Nature Communications.

References:

1. Maze, J. R. et al. Nanoscale magnetic sensing with an individual electronic spin in diamond. *Nature* 455, 644–647 (2008).
2. Budker, D. A new spin on magnetometry. *Nature(London)* 422, 574–575 (2003).
3. Budker, D. & Romalis, M. Optical magnetometry. *Nat. Phys.* 3, 227–234 (2007).
4. Taylor, J. M. et al. High-sensitivity diamond magnetometer with nanoscale resolution. *Nat. Phys.* 4, 810–816 (2008).
5. Kominis, I. K., Kornack, T. W., Allred, J. C. & Romalis, M. V. A subfemtotesla multichannel atomic magnetometer. *Nature* 422, 596–599 (2003).
6. Auzinsh, M. et al. Can a quantum nondemolition measurement improve the sensitivity of an atomic magnetometer? *Phys. Rev. Lett.* 93, 173002–173002 (2004).
7. Savukov, I. M., Seltzer, S. J., Romalis, M. V. & Sauer, K. L. Tunable Atomic Magnetometer for Detection of Radio-Frequency Magnetic Fields. *Phys. Rev. Lett.* 95, 063004 (2005).
8. Herrera-Martí, D. A., Gefen, T., Aharonov, D., Katz, N. & Retzker, A. Quantum Error-Correction-Enhanced Magnetometer Overcoming the Limit Imposed by Relaxation. *Phys. Rev. Lett.* 115, 200501 (2015).
9. Kominis, I. K. Sub-Shot-Noise Magnetometry with a Correlated Spin-Relaxation Dominated Alkali-Metal Vapor. *Phys. Rev. Lett.* 100, 073002 (2008).
10. Brask, J. B., Chaves, R. & Kołodyński, J. Improved Quantum Magnetometry beyond the Standard Quantum Limit. *Phys. Rev. X* 5, 031010 (2015).

Reviewer #2

The authors have reviewed and addressed all of my previous concerns. Specifically, they now explain that their method is robust to realistic variations of the system parameters while still retaining the advantages for metrology of magnetic fields. The idea of using a dynamical system which is not integrable as a sensor is very interesting and I believe the authors' analysis show in principle that it could eventually be applied in reality. With all the changes and the new plots, I think this paper should be accepted for publication.

We are pleased that Reviewer #2 recommends publication in Nature Communications and thank him/her for his/her constructive criticism that has helped improve the paper.

Reviewer #3

In summary, I am basically satisfied with the authors' response to my report. They note that that they have chosen to present a fairly specific model as an example of their general point, and I am happy with this as a response, even though it is a bit different to what I had written (in fact I think it is better).

As such I am happy to say that I think the manuscript in its present form is publishable in Nature Communications.

Just a very slight note --- in the abstract it should be "kicks render" rather than "kicks renders."

We are happy that also Reviewer #3 recommends publication in Nature Communications and thank him/her for his/her constructive criticism that has helped improve the paper. We implemented the correction of the typo in the abstract.

Changes of the manuscript compared to the last version:

We profited from the time to improve the manuscript even further. In fact, we realized that Doppler- and pressure broadening, which are usually neglected in the simulation of SERF magnetometers, leads to additional decoherence in the excited state used for the kicking, such that the previously considered SERF regime is not optimal when including kicking. In the new simulations we take the Doppler broadening into account, and find that in another parameter regime (lower density of the atomic vapor where pressure broadening is negligible, lower temperatures, and even smaller magnetic fields) the gain due to the non-linear kicking is even more substantial than reported in the last version of the manuscript: an improvement of 68% in sensitivity compared to the previous 20%. Again, the simulations and parameters are based largely on an existing experiment (see [5] in Supplemental Material) and are therefore highly relevant for further improving the best existing magnetometers. We therefore prefer to include this example rather than the last one in the paper.

Besides this change, resulting in a new fig. 7 with more impressive improvement, and the modified numerical values plus inclusion of Doppler broadening for the simulation, we left the paper essentially unchanged. All changes are documented in color in the file "QChaoticSensor_changes_v2.pdf".

REVIEWERS' COMMENTS:

Reviewer #2 (Remarks to the Author):

I have reviewed the new version of the manuscript and the point by point response to the referees. I agree with the authors in regard to their response to Referee #1. The present manuscript is about enhanced magnetometry using a quantum chaotic dynamical system and not about measuring the state of a qubit. From referee #1's response alone, it is not clear to me what the problem and/or where the source of confusion is. As such, and given the continuous improvement in the manuscript, I still recommend the paper to be published in Nature Communications.

We thank Reviewer #2 for recommending the paper to be published in Nature Communications.